# A New Proof for a Result on the Inclusion Chromatic Index of Subcubic Graphs

**Lily Chen *** and **Yanyi Li**

School of Mathematical Sciences, Huaqiao University, Quanzhou 362000, China; liyanyi0708@163.com
* Correspondence: lily60612@126.com

**Abstract:** Let $G$ be a graph with a minimum degree $\delta$ of at least two. The inclusion chromatic index of $G$, denoted by $\chi'_\subset(G)$, is the minimum number of colors needed to properly color the edges of $G$ so that the set of colors incident with any vertex is not contained in the set of colors incident to any of its neighbors. We prove that every connected subcubic graph $G$ with $\delta(G) \geq 2$ either has an inclusion chromatic index of at most six, or $G$ is isomorphic to $\hat{K}_{2,3}$, where its inclusion chromatic index is seven.

**Keywords:** inclusion-free edge coloring; subcubic; adjacent-vertex-distinguishing edge coloring

## 1. Introduction

Graph coloring is an abstraction for partitioning a set of binary-related objects into subsets of independent objects; it has many practical applications [1]. The chromatic index and chromatic polynomials are two important parameters in graph theory. There are also many chemical applications to the chromatic index and chromatic polynomials; see ([2–8]). In this paper, we will study an edge coloring: inclusion-free edge coloring. Graphs in this article are assumed to be simple and undirected. Let $G$ be a graph with minimum degree $\delta \geq 2$ and let $\phi$ be a proper edge coloring of $G$. For every $v \in V(G)$, the *palette* of $v$ is defined to be

$$S_\phi(v) = \{\phi(e) | e \text{ is incident to } v\}.$$

The *inclusion-free edge coloring*, recently introduced by Przybyłło and Kwaśny [9], is a proper edge coloring $\phi$ of $G$ such that for every $uv \in E(G)$, neither $S_\phi(u) \subseteq S_\phi(v)$ nor $S_\phi(v) \subseteq S_\phi(u)$. The requirement of $\delta \geq 2$ is necessary since the palette of a degree-1 vertex is always a subset of the palette of its unique neighbor. The minimum number of colors required in an inclusion-free edge coloring of $G$ is called the *inclusion chromatic index* and is denoted by $\chi'_\subset(G)$.

Actually, the concept of the inclusion-free edge coloring was first introduced by Zhang [10], where it was named as Smarandachely adjacent vertex edge coloring. Then, Gu et al. [11] also investigated the topic and named the coloring as strict neighbor-distinguishing edge coloring. Although their names are different, they were all introduced to strengthen the adjacent-vertex-distinguishing edge coloring, or for short, AVD-edge coloring. An *AVD-edge coloring* of $G$ is a proper edge coloring $\phi$ such that for every $uv \in E(G)$, $S_\phi(u) \neq S_\phi(v)$; the minimum number of colors needed in an AVD-edge coloring is called the *AVD chromatic index*, denoted by $\chi'_a(G)$. Clearly a graph $G$ has an AVD-edge coloring if and only if $G$ contains no isolated edges. Note that for a regular graph $G$, the palettes of any two vertices are different if and only if neither is contained in the other; hence, $\chi'_a(G) = \chi'_\subset(G)$.

The AVD-edge coloring has attracted the attention of several groups of graph theorists. It was conjectured by Zhang et al. [12] that $\chi'_a(G) \leq \Delta + 2$ for any connected graph $G$ with $|V(G)| \geq 3$ that is not the cycle $C_5$. Balister et al. [13] proved that the conjecture holds for the class of bipartite graphs and for the class of subcubic graphs; they also showed

that in general, $\chi'_a(G) \leq \Delta + O(\log \chi(G))$ where $\chi(G)$ is the chromatic number of $G$. More recently, Hatami [14] showed that $\chi'_a(G) \leq \Delta + 300$.

Despite the similarity between the two invariant $\chi'_a(G)$ and $\chi'_\subset(G)$, the upper bound for $\chi'_\subset(G)$ seems to be much larger than that of $\chi'_a(G)$. Przybyłło and Kwaśny [9] showed that if $G$ is a complete bipartite graph, then $\chi'_\subset(G) = (1 + \frac{1}{\delta-1})\Delta$, where $\delta$ is the minimum degree of $G$. By using a greedy coloring scheme, they showed that in general $\chi'_\subset(G) \leq 3\Delta - 1$ where $\Delta$ is the maximum degree of $G$. They made the following conjecture:

**Conjecture 1.** *Let $G$ be a connected graph with minimum degree $\delta \geq 2$ and maximum degree $\Delta$ that is not isomorphic to $C_5$. Then*

$$\chi'_\subset(G) \leq \lceil (1 + \frac{1}{\delta-1})\Delta \rceil.$$

Using a probabilistic approach, Przybyłło and Kwaśny [9] proved the following upper bound for $\chi'_\subset(G)$, which is not as strong as the conjectured bound in Conjecture 1.

**Theorem 1.** *If $G$ is a graph with minimum degree $\delta \geq 2$ and maximum degree $\Delta$, then*

$$\chi'_\subset(G) \leq (1 + \frac{4}{\delta})\Delta + O(\Delta^{\frac{2}{3}} \log^4 \Delta).$$

It turns out that there exists a class of exceptional graphs to Conjecture 1 in the case of $\delta = 2$: for $\Delta \geq 3$, let $\hat{K}_{2,\Delta}$ be the graph obtained from the complete bipartite graph $K_{2,\Delta}$ by subdividing an edge exactly once; see Figure 1. It is easy to check that no two edges of $\hat{K}_{2,\Delta}$ can receive the same color in an inclusion-free edge coloring; hence, $\chi'_\subset(\hat{K}_{2,\Delta}) = 2\Delta + 1$, which is the number of edges of $\hat{K}_{2,\Delta}$.

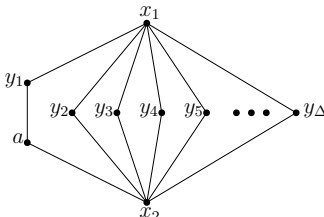

**Figure 1.** The graph $\hat{K}_{2,\Delta}$.

We strongly believe that $\hat{K}_{2,\Delta}$ may be the only exception to Conjecture 1. So Conjecture 1 needs to be slightly modified by adding the condition that $G$ is not isomorphic to $\hat{K}_{2,\Delta}$. Gu et al. [11] confirmed the modified conjecture for the class of subcubic graphs. A graph $G$ is *formal* if $\delta(G) \geq 2$. They proved the following result:

**Theorem 2.** *Let $G$ be a connected formal subcubic graph. Then $\chi'_\subset(G) \leq 7$, and moreover, $\chi'_\subset(G) = 7$ if and only if $G$ is isomorphic to the graph $\hat{K}_{2,3}$.*

They proved the result by contradiction. Let $G$ be a counterexample with a minimal number of edges, by establishing a series of auxiliary claims, they showed that $G$ does not contain a 2-vertex adjacent to two 2-vertices, and any 3-vertex of $G$ cannot be adjacent to a 2-vertex, that is, $G$ must be 3-regular, and hence, $\chi'_\subset(G) \leq 5$, a contradiction.

In this paper, we will give a shorter proof of this theorem. We also prove the result by contradiction. First, we establish a lemma for forbidden colors and use it to exclude some structures. We also show that $G$ does not contain a 2-vertex adjacent to two 2-vertices, i.e., $G$ contains no $k$-thread with $k \geq 3$, and $G$ does not contain a 3-cycle with one 2-vertex, and a 4-cycle with two non-adjacent 2-vertices. Then, we show that if $G$ contains a 1-thread or 2-thread, it must be isomorphic to $\hat{K}_{2,3}$.

## 2. Proof of the Main Result

Let $G$ be a connected subcubic graph with $\delta(G) = 2$. Suppose that $\chi'_C(G) \geq 7$. We pick a graph $G$ such that $|V(G)| + |E(G)|$ is as small as possible. By a *good coloring*, we mean an inclusion-free edge coloring using at most six colors. If $G$ and $H$ are two graphs with $|E(H)| + |V(H)| < |E(G)| + |V(G)|$, we will say that $H$ is *smaller* than $G$. We will show that $G$ is isomorphic to $\hat{K}_{2,3}$.

Let $C = \{1, 2, 3, 4, 5, 6\}$ be a set of six colors. Suppose that $\phi$ is a good coloring of a proper subgraph $G'$ of $G$ using colors from $C$. Let $e = uv$ be an edge in $E(G) \backslash E(G')$. We denote by $A_\phi(e)$ the set of colors that are available for $e$. To color $e$, one cannot use a color from $S_\phi(u)$; moreover, for each neighbor $v'$ of $u$ other than $v$, if by assigning a color $\alpha$ to $e$, we would have either $S_\phi(u) \subseteq S_\phi(v')$ or $S_\phi(v') \subseteq S_\phi(u)$, then the color $\alpha$ cannot be used for $e$. We call these two types of colors *the forbidden colors of e by the vertex u*, denoted by $F_\phi(e, u)$. It follows that $A_\phi(e) = C \backslash (F_\phi(e, u) \cup F_\phi(e, v))$.

For simplicity, we use *k-vertex* to denote a vertex with degree $k$. Similarly, by *k-neighbor* of a vertex $u$, we mean a neighbor of $u$ that has degree $k$.

**Lemma 1.** *Suppose that $G'$ is a proper subgraph of $G$ with $\delta(G') = 2$ and that $\phi$ is a good coloring of $G'$. Let $e = uv$ be an edge in $E(G) \backslash E(G')$, where $u$ is a 2-vertex of $G'$. Then*

- $|F_\phi(e, u)| = 2$ *if both neighbors of $u$ in $G'$ are 3-vertices;*
- $|F_\phi(e, u)| = 3$ *if exactly one neighbor of $u$ in $G'$ is a 3-vertex;*
- $|F_\phi(e, u)| \leq 4$ *if both neighbors of $u$ in $G'$ are 2-vertices.*

**Proof.** Let $v'$ and $v''$ be the two neighbors of $u$ in $G'$. Since $\phi$ is a good coloring of $G'$, $\phi(uv'') \notin S_\phi(v')$. Therefore, no matter what color we assign to the edge $uv$, we will have that $S_\phi(u) \not\subseteq S_\phi(v')$. Now if $v'$ is a 3-vertex of $G'$, then the only color in $S_\phi(v')$ that is forbidden for $e$ is $\phi(uv')$; while if $v'$ is a 2-vertex of $G'$, then neither color in $S_\phi(v')$ can be used for $e$ since we require $S_\phi(v') \not\subseteq S_\phi(u)$. By symmetry, the same holds for $v''$. So Lemma 1 follows immediately. (Note that we may have that $|F_\phi(e, u)| = 3$ in the case of $d_{G'}(v') = d_{G'}(v'') = 2$: this happens when $S_\phi(v') \cap S_\phi(v'') \neq \emptyset$.)  $\square$

Actually, Lemma 1 can be extended to more general situations: let $G$ be a connected graph with $\delta(G) \geq 2$. Suppose that $G'$ is a proper subgraph of $G$ with $\delta(G') \geq 2$ and that $\phi$ is an inclusion-free edge coloring of $G'$. Let $e = uv$ be an edge in $E(G) \backslash E(G')$. Then $|F_\phi(e, u)| \leq d_u + N_u$, where $d_u$ is the degree of $u$ in $G'$, and $N_u$ is the number of neighbors of $u$ in $G'$ with degree no more than $d_u$.

For integer $k \geq 0$, a *k-thread* of $G$ is a path $P = v_0 v_1 v_2 \cdots v_{k+1}$ of length $k + 1$ such that both $v_0$ and $v_{k+1}$ are 3-vertices, and each of $v_1, v_2, \cdots, v_k$ is a 2-vertex. So a 0-thread is an edge that is incident to two 3-vertices. A *k-thread* $P$ is called *separating* if deleting all the internal vertices in $P$ yields a disconnected subgraph of $G$.

**Lemma 2.** *$G$ contains no separating k-thread for $k \geq 0$.*

**Proof.** Let $P = v_0 v_1 v_2 \cdots v_{k+1}$ be a separating *k*-thread and let $G'$ be the subgraph of $G$ obtained by deleting all the internal vertices in $P$. Since $G'$ is disconnected, we assume that $G_1$ and $G_2$ are the two components of $G'$ with $v_0 \in V(G_1)$ and $v_{k+1} \in V(G_2)$. Since $G'$ is a proper subgraph of $G$, $G'$ has a good coloring $\phi$. We will extend $\phi$ to $G$ by assigning colors to the edges on the thread P.

First we assume that $k \geq 1$. By permuting colors in $G_1$ if necessary, we may assume that $S_\phi(v_0) = S_\phi(v_{k+1})$. Clearly $v_0$ is a 2-vertex in $G'$. By Lemma 1, $|F_\phi(v_0 v_1, v_0)| \leq 4$, and hence, $|A_\phi(v_0 v_1)| \geq 2$. By symmetry, $|A_\phi(v_k v_{k+1})| \geq 2$. So we may assign distinct colors to $v_0 v_1$ and $v_k v_{k+1}$, then color all other edges on the thread one by one in the following order: $v_1 v_2, v_2 v_3, \cdots, v_{k-1} v_k$. Note that in each step, the edge to be colored forbids at most five colors, and hence, it has at least one color available. So we obtain a good coloring of $G$.

Next assume that $k = 0$, i.e., $G$ has a cut edge that is incident to two 3-vertices. Then we can permute colors in $G_1$ so that $F_\phi(v_0v_1, v_1) \subseteq F_\phi(v_0v_1, v_0)$ if $|F_\phi(v_0v_1, v_1)| \leq |F_\phi(v_0v_1, v_0)|$ or $F_\phi(v_0v_1, v_0) \subseteq F_\phi(v_0v_1, v_1)$ otherwise; hence, there are at least two colors available for $v_0v_1$ and it can be colored.

In each case, we obtain a good coloring of $G$, contrary to our assumption. Therefore, $G$ contains no separating $k$-thread for $k \geq 0$. $\square$

For general case, suppose that $G$ is a connected graph with $\delta(G) \geq 2$, $P = v_0v_1v_2 \cdots v_{k+1}$ is a separating $k$-thread in $G$. Let $G'$ be the graph obtained by deleting all the internal vertices of $P$, and $G_1$, $G_2$ be the two components of $G'$. By the similar proof as Lemma 2, we have $\chi'_C(G) \leq max\{\chi'_C(G_1), \chi'_C(G_2), |F_\phi(v_0v_1, v_0)|, |F_\phi(v_kv_{k+1}, v_{k+1})|\} + 3$. Since $|F_\phi(e, u)| \leq d_u + N_u \leq 2d_u$, $\chi'_C(G) \leq max\{\chi'_C(G_1), \chi'_C(G_2), 2d_{v_0}, 2d_{v_{k+1}}|\} + 3$.

**Lemma 3.** *Let $G'$ be a subgraph of $G$ with $\delta(G') \geq 2$. Suppose that $P = v_0v_1v_2 \cdots v_{k+1}$ is a $k$-thread in $G'$ with $k \geq 3$, then $G'$ has a good coloring $\phi$ such that $\phi(v_0v_1) = \phi(v_kv_{k+1})$. In particular, $G$ contains no $k$-thread with $k \geq 3$.*

**Proof.** First assume that $v_0$ is not adjacent to $v_{k+1}$. Then let $G''$ be the graph obtained by adding the edge $v_0v_{k+1}$ to $G' \backslash \{v_1, v_2, \cdots v_k\}$. Clearly $\delta(G'') \geq 2$. So $G''$ has a good coloring $\phi'$. We can construct a good coloring $\phi$ of $G'$ as follows: $\phi(v_0v_1) = \phi(v_kv_{k+1}) = \phi'(v_0v_{k+1})$; $\phi(e) = \phi'(e)$ for all $e \in E(G') \cap E(G'')$. We color the remaining edges in the following order: $v_1v_2, v_2v_3, \cdots, v_{k-1}v_k$. Since $k \geq 3$, at each step, the edge to be colored forbids at most five colors. Therefore, all edges of $P$ can be colored and we obtain a good coloring $\phi$ of $G'$ with $\phi(v_0v_1) = \phi(v_kv_{k+1})$.

Next assume that $v_0$ is adjacent to $v_{k+1}$. Let $G'' = G' \backslash \{v_1, v_2, \cdots v_k\}$ and let $\phi$ be a good coloring of $G''$. Then each of $A_\phi(v_0v_1)$ and $A_\phi(v_kv_{k+1})$ has size at least three. Since $\phi(v_0v_{k+1}) \notin A_\phi(v_0v_1) \cup A_\phi(v_kv_{k+1})$. We have that $A_\phi(v_0v_1) \cap A_\phi(v_kv_{k+1}) \neq \varnothing$. We may pick $\alpha \in A_\phi(v_0v_1) \cap A_\phi(v_kv_{k+1})$ and assign it to $v_0v_1$ and $v_kv_{k+1}$. Similar as above, the remaining edges of $P$ can be colored in the order: $v_1v_2, v_2v_3, \cdots, v_{k-1}v_k$. Therefore, $G'$ has a good coloring $\phi$ such that $\phi(v_0v_1) = \phi(v_kv_{k+1})$.

In particular, if $G' = G$, and $G$ has a $k$-thread with $k \geq 3$, then $G$ has good coloring $\phi$, contrary to our assumption. Hence, $G$ contains no $k$-thread with $k \geq 3$. $\square$

Lemma 3 can also be extended to more general situations: let $G$ be a connected graph with $\delta(G) \geq 2$, and $H$ be a graph obtained from $G$ by subdividing an edge with at least 3 vertices, then $\chi'_C(H) \leq max\{\chi'_C(G), 6\}$.

**Lemma 4.** *Let $P$ be a 1- or 2-thread in $G$. Then the two 3-vertices on $P$ are not adjacent to each other.*

**Proof.** Suppose that $P = uwv$ is a 1-thread in $G$ where $u$ is adjacent to $v$. Let $u'$ (resp. $v'$) be the neighbor of $u$ (resp. $v$) not on $P$. Note that $G' = G \backslash w$ is a subcubic graph with minimum degree 2. By our assumption on $G$, $G'$ has good coloring $\phi$. Since $d_{G'}(u) = d_{G'}(v) = 2$, $\phi(uv) \notin S_\phi(u')$ and $\phi(uv) \notin S_\phi(v')$. It is easy to see that if $u'$ is a 3-vertex, $|A_\phi(uw)| \geq 3$ and if $u'$ is a 2-vertex, $|A_\phi(uw) \geq 2$. By symmetry, $|A_\phi(vw)| \geq 2$. So we can assign two distinct colors to $uw$ and $vw$ to obtain a good coloring of $G$, a contradiction.

The case when $P$ is a 2-thread can be proved in a similar manner. $\square$

**Lemma 5.** *Let $uvxyu$ be a 4-cycle of $G$. If $d_G(u) = d_G(x) = 3$ and $d_G(v) = d_G(y) = 2$, then $G$ is isomorphic to $\hat{K}_{2,3}$.*

**Proof.** Let $u'$ (resp. $x'$) be the neighbor of $u$ (resp. $x$) other than $v$ and $y$.

First we assume that $u' = x'$. In this case, if $d_G(u') = 2$, then $G \cong K_{2,3}$, contrary to our assumption that $\chi'_C(G) \geq 7$. If $d_G(u') = 3$, let $w$ be the neighbor of $u'$ other than $u, x$, then the edge $u'w$ lies in a separating $k$-thread with $k \geq 0$, contrary to Lemma 2. Hence, $u' \neq x'$.

Let $\phi$ be a good coloring of $G' = G\backslash v$. Then $A_\phi(uv) = C\backslash(F_\phi(uv, u) \cup S_\phi(x))$. So if one of $u'$ and $x'$ is a 3-vertex, then by Lemma 1, one of $uv$ and $vx$ has at least two colors available and the other one has at least one color available. So they can both be colored. Therefore, $d_G(x') = d_G(u') = 2$.

Note that if $u'$ is adjacent to $x'$, then $G \cong \hat{K}_{2,3}$. So we may assume that $u'$ is not adjacent to $x'$. Let $G'$ be the graph obtained by adding the edge $u'x'$ in $G\backslash\{u, v, x, y\}$. Clearly $\delta(G') \geq 2$. So $G'$ has a good coloring $\phi'$. Since $d_{G'}(u') = d_{G'}(x') = 2$, the edges $u'u''$, $u'x'$, $x'x''$ receive different colors, where $u''$ (resp. $x''$) be the neighbor of $u'$ (resp. $x'$). We may assume that $\phi(u'u'') = 1$, $\phi(u'x') = 2$, $\phi(x'x'') = 3$, then we color the edges $uu', uv, uy, vx, xy, xx'$ as follows: $\phi(uu') = \phi(xx') = 2$, $\phi(uv) = 3$, $\phi(vx) = 4$, $\phi(xy) = 5$, $\phi(uy) = 6$. It is easy to see that this coloring is a good coloring of $G$, contrary to our assumption. □

Recall that Balister et al. [13] showed that a 3-regular graph has an AVD chromatic index of at most 5. Since the inclusion chromatic index is the same as the AVD chromatic index for regular graphs, every 3-regular graph has an inclusion chromatic index of at most 5. Since $\chi'_\subset(G) \geq 7$ by our assumption, $G$ must have at least one 2-vertex. By Lemma 3, $G$ contains either a 1-thread or a 2-thread. Let $P$ be a $k$-thread with $k = 1$ or 2, and let $G'$ be the graph obtained from $G$ by deleting all internal vertices of $P$. By Lemma 2, $G'$ is connected. Clearly, $G'$ is a subcubic graph with minimum degree 2 and is smaller than $G$. By our assumption on $G$, $G'$ has a good coloring $\phi$. We will extend $\phi$ to a good coloring of $G$ by assigning appropriate colors for all edges on the thread $P$.

**Lemma 6.** *If $P$ is a 2-thread in $G$, then $G$ is isomorphic to $\hat{K}_{2,3}$.*

**Proof.** Suppose that $P = uu'v'v$ is a 2-thread where $d_G(u') = d_G(v') = 2$ and $d_G(u) = d_G(v) = 3$. By Lemma 2, $u \neq v$, and by Lemma 4, $u$ is not adjacent to $v$. Let $u_1$ and $u_2$ be the neighbors of $u$ other than $u'$ and let $v_1$ and $v_2$ be the neighbors of $v$ other than $v'$.

Note that the edge $uu'$ can be colored by any color not in $F_\phi(uu', u)$. By Lemma 1, $|A_\phi(uu')| \geq 2$; by symmetry, $|A_\phi(vv')| \geq 2$. The edge $u'v'$ can be colored by any color not in $S_\phi(u) \cup S_\phi(v)$, so $|A_\phi(u'v')| \geq 2$.

Assume that there exists a 3-vertex in $\{u_1, u_2, v_1, v_2\}$, say $u_1$. Then by Lemma 1, the edge $uu'$ forbids at most three colors, and hence, the edges on $P$ can be colored in the order of $vv'$, $u'v'$, and $uu'$. We obtain a good coloring of $G$, a contradiction.

Therefore, we have that $d_G(u_1) = d_G(u_2) = d_G(v_1) = d_G(v_2) = 2$. Note that if $\{u_1, u_2\} = \{v_1, v_2\}$, then $G$ is isomorphic to $\hat{K}_{2,3}$. So we may assume that $|\{u_1, u_2\} \cap \{v_1, v_2\}| \leq 1$.

Case 1: $|\{u_1, u_2\} \cap \{v_1, v_2\}| = 1$.

By symmetry, assume that $u_1 = v_1$. Then, $uu'v'vu_1u$ form a 5-cycle, call it $C_1$. Let $G''$ be the graph obtained from $G\backslash\{u', v', u_1\}$ by identifying $u$ and $v$. Let $w$ be the new identified vertex, and let $u'_2$ (resp $v'_2$) be the neighbor of $u_2$ in $G''$ other than $w$. Clearly, $G''$ is a subcubic graph with minimum degree 2 and is smaller than $G$. So $G''$ has a good coloring $\psi'$. We extend $\psi'$ to a good coloring $\psi$ of $G$ as follows: let $\psi(uu_2) = \psi'(wu_2)$, $\psi(vv_2) = \psi'(wv_2)$, and $\psi(e) = \psi'(e)$ for $e \in E(G) \cap E(G'')$. Now we need to assign colors to edges on $C_1$: Since $\psi'$ is a good coloring of $G''$, among the four edges $uu_2, u_2u'_2, vv_2$ and $v_2v'_2$, only $u_2u'_2$ and $v_2v'_2$ may share a same color. So we may assume that $\psi(uu_2) = 1$, $\psi(vv_2) = 2$, $\psi(u_2u'_2) = 3$, and $\psi(v_2v'_2) = 3$ or 4. In both cases, we will set $\psi(uu') = 2$, $\psi(vv') = 1$, $\psi(uu_1) = 4$, $\psi(vv_1) = 5$, and $\psi(u'v') = 6$. It is easy to check that $\psi$ is a good coloring of $G$, a contradiction.

Case 2: $N(u) \cap N(v) = \emptyset$.

For $x \in \{u, v\}$ and $i \in \{1, 2\}$, let $x'_i$ be the neighbor of $x_i$ other than $x$. By Lemmas 5 and 2, $u'_1 \neq u'_2$ and $v'_1 \neq v'_2$. If $u'$ is adjacent to $u'_2$, then by Lemma 2, one of them must have degree 3, say $d_G(u'_1) = 3$. We claim that $d_G(u'_2) = 3$, as, otherwise, this can be reduced to Case 1 by choosing the 2-thread $uu_2u'_2u'_1$ to begin with. By Lemma 3, we may choose

$\phi$ such that $\phi(u_1u'_1) = \phi(u_2u'_2)$. Note that the edge $uu'$ can be assigned any color not in $S_\phi(u_1) \cup S_\phi(u_2)$; so $|A_\phi(uu')| \geq 3$. Similarly, $|A_\phi(u'v')| \geq 2$ and $|A_\phi(vv')| \geq 2$. So the edges $vv'$, $u'v'$, and $uu'$ can be colored in that order. $\square$

**Lemma 7.** *Let $uwvu_1u$ be a 4-cycle of G with $d(u) = d(v) = d(u_1) = 3$ and $d(w) = 2$, and let $u_2$ (resp. $v_2$) be the neighbor of u (resp. v) other than w and $u_1$. If $d(u_2) = d(v_2) = 2$, then the graph $G' = G\backslash w$ has a good coloring $\phi$ so that $\phi(uu_2) = \phi(vv_2)$.*

**Proof.** Let $u'_1$ be the neighbor of $u_1$ other than $u$ and $v$ and let $u'_2$ (resp. $v'_2$) be the neighbor of $u_2$ (resp. $v_2$) other than $u$ (resp. $v$). By Lemma 2, $u_2v_2 \notin E(G)$. Since $G'$ is a subcubic graph with minimum degree 2 and is smaller than $G$, $G'$ has a good coloring $\phi$. Now we remove the colors on the edges $u_2u'_2$, $uu_2$, $uu_1$, $vu_1$, $vv_2$, and $v_2v'_2$. Then $|A_\phi(uu_2)| \geq 3$, and $|A_\phi(vv_2)| \geq 3$. Note that $|A_\phi(uu_2) \cup A_\phi(vv_2)| \leq 5$ since $\phi(u_1u'_1) \notin A_\phi(uu_2) \cup A_\phi(vv_2)$. So $A_\phi(uu_2) \cap A_\phi(vv_2) \neq \varnothing$. Choose a color $\alpha \in A_\phi(uu_2) \cap A_\phi(vv_2)$ and assign it to edges $uu_2$ and $vv_2$.

If either $u'_2 = v'_2$ or $u'_2$ is adjacent to $v'_2$, then each of $u_2u'_2$ and $v_2v'_2$ has at least two colors available. So we will color them using different colors. Now each of $uu_1$ and $vu_1$ has at least two colors available, so they can be colored as well. So we may assume that neither $u'_2 = v'_2$ nor $u'_2$ is adjacent to $v'_2$, and hence, $u_2u'_2$ and $v_2v'_2$ may receive the same color.

Now we have that $|A_\phi(u_2u'_2)| \geq 1$, $|A_\phi(uu_1)| \geq 3$, and $|A_\phi(vu_1)| \geq 3$ and $|A_\phi(v_2v'_2)| \geq 1$. We then color $u_2u'_2$ and $v_2v'_2$ independently. The edge $uu_1$ (resp. $vv_1$) may only lose the color assigned to $u_2u'_2$ (resp. $v_2v'_2$). So both $uu_1$ and $vv_1$ still have at least two colors available, and hence, they can be colored. $\square$

Finally we consider the case that $P$ is a 1-thread.

**Lemma 8.** *If $P$ is a 1-thread in G, then G is isomorphic to $\hat{K}_{2,3}$.*

**Proof.** Let $P = uwv$. Then by Lemma 4, $u$ is not adjacent to $v$. Let $u_1$, $u_2$ be the neighbors of $u$ other than $w$, and let $v_1$, $v_2$ be the neighbors of $v$ other than $w$. We consider the following three cases.

Case 1: $\{u_1, u_2\} = \{v_1, v_2\}$.

Assume that $u_1 = v_1$ and $u_2 = v_2$. By Lemma 5, neither $u_1$ nor $u_2$ is a 2-vertex. So $d_G(u_1) = d_G(u_2) = 3$. By Lemma 1, each of $uw$ and $vw$ forbids at most four colors. So they both can be colored.

Case 2: $|\{u_1, u_2\} \cap \{v_1, v_2\}| = 1$

Suppose that $u_1 = v_1$ and $u_2 \neq v_2$. By Lemma 5, $d_G(u_1) = 3$. Note that the edge $uw$ can be assigned any color not in $F_\phi(uw, u) \cup S_\phi(v)$ and the edge $vw$ can be assigned any color not in $F_\phi(vw, v) \cup S_\phi(u)$. So if one of $u_2$ and $v_2$ is a 3-vertex, then by Lemma 1, one of $uw$ and $vw$ has at least two colors available, while the other one has at least one color available. So we can extend $\phi$ to a good coloring of $G$, a contradiction.

Therefore, we may assume that $d_G(u_2) = d_G(v_2) = 2$. Let $u'_1$ be the neighbor or $u_1$ other than $u$ and $v$ and let $u'_2$ (resp. $v'_2$) be the neighbors of $u_2$ (resp. $v_2$) other than $u$ (resp. $v$). By Lemma 7, the graph $G' = G\backslash w$ has a good coloring $\phi$ so that $\phi(uu_2) = \phi(vv_2)$. It is easy to see that $|A_\phi(uw)| \geq 2$, and $|A_\phi(vw)| \geq 2$. Therefore, we may extend $\phi$ to a good coloring of $G$, a contradiction.

Case 3: $\{u_1, u_2\} \cap \{v_1, v_2\} = \varnothing$

Note that $A_\phi(uw) = C\backslash(F_\phi(uw, u) \cup S_\phi(v))$ and $A_\phi(vw) = C\backslash(F_\phi(vw, v) \cup S_\phi(u))$. Therefore, if at least three of $u_1$, $u_2$, $v_1$, and $v_2$ are 3-vertices, then by Lemma 1, one of the edges $uw$ and $vw$ has at least two colors available, while the other one has at least one color available. So we can extend $\phi$ to a good coloring of $G$.

Therefore, at most, two of the vertices in $\{u_1, u_2, v_1, v_2\}$ are 3-vertices. For $i \in \{1, 2\}$ we will use $u'_i$ (resp. $v'_i$) to denote a neighbor of $u_i$ (resp. $v_i$) different from $u$ (resp. $v$). By symmetry, it suffices to consider the following two subcases:

Subcase 3.1: both $u_1$ and $u_2$ are 2-vertices.

Since $G$ contains no 2-thread, each of $u_1'$ and $u_2'$ is a 3-vertex. By Lemmas 2 and 5, $u_1' \neq u_2'$. So by Lemma 3, we can choose a good coloring $\phi$ of $G\backslash w$ with $\phi(u_1 u_1') = \phi(u_2 u_2')$. Then the edge $uw$ has at least one color available. If the edge $vw$ has at least two colors available, then $\phi$ can be extended to a good coloring of $G$. Therefore, at least one of $v_1$ and $v_2$ is a 2-vertex, say $v_1$. Moreover, if $S_\phi(u) \cap S_\phi(v) \neq \varnothing$, then one of $uw$ and $vw$ has two available colors, while the other one has at least one available color, so $\phi$ can be extended to a good coloring of $G$.

So we may assume that $\phi(uu_1) = 1$, $\phi(uu_2) = 2$, $\phi(vv_1) = 3$, $\phi(vv_2) = 4$, and $\phi(u_1 u_1') = \phi(u_2 u_2') = 5$. If the color $5 \notin S_\phi(v_1) \cup S_\phi(v_2)$, or $d_G(v_2) = 3$ and $5 \notin S_\phi(v_1)$, then we may assign color 5 to $vw$ and assign color 6 to $uw$ to obtain a good coloring of $G$. So we can assume that $\phi(v_1 v_1') = 5$.

Observe that if $\{3, 4\} \nsubseteq S_\phi(u_1')$, say $3 \notin S_\phi(u_1')$, then by changing the color of $uu_1$ from 1 to 3, we obtain that $|A_\phi(uw)| \geq 2$ and $|A_\phi(vw)| \geq 1$. So we can extend $\phi$ to a good coloring of $G$. So we have that $S_\phi(u_1') = \{3, 4, 5\}$. Similarly $S_\phi(u_2') = \{3, 4, 5\}$.

Next we will show that $v_2$ must be a 2-vertex. Assume that $d_G(v_2) = 3$. Note that the color $3 \notin S_\phi(v_2)$ since $\phi$ is a good coloring of $G'$. So if $\{1, 2\} \nsubseteq S_\phi(v_1')$, say $1 \notin S_{\phi'}(v_1')$, then we may change the color of $vv_1$ from 3 to 1, assign color 3 to $vw$ and color 6 to $uw$; we obtain a good coloring of $G$. So $S_\phi(v_1') = \{1, 2, 5\}$. Now we can change the colors of $uu_1$ and $vv_1$ both to 6, and let $\phi(uw) = 1$ and $\phi(vw) = 3$, we obtain a good coloring of $G$.

Therefore, we know that $d_G(v_2) = 2$. Observe that $v_2'$ is a 3-vertex. If $S_\phi(v_2') \neq \{1, 2, 6\}$, then we can pick a color $\beta \in \{1, 2, 6\}\backslash S_{\phi'}(v_2')$ and change the color of $vv_2$ from 4 to $\beta$; if $\beta = 6$, we will also change the color of $uu_1$ from 1 to 6. Now we have $S_\phi(u) \cap S_\phi(v) \neq \varnothing$, so we can extend $\phi$ to a good coloring of $G$.

Therefore, we have that $S_\phi(v_2') = \{1, 2, 6\}$. We construct a good coloring $\phi'$ of $G' = G\backslash w$ as follows: for all $e \in E(G')\backslash\{vv_1, vv_2, v_2 v_2'\}$, let $\phi'(e) = \phi(e)$; for the edge $v_2 v_2'$, note that $|A_{\phi'}(v_2 v_2')| \geq 2$. So we can set $\phi'(v_2 v_2') \neq \phi(v_2 v_2')$. Each of $vv_1$ and $vv_2$ has at least two colors available, so they can both be colored. In the new coloring $\phi'$, since $|S_{\phi'}(v_2)\backslash S_\phi(v_2)| = 1$, $S_{\phi'}(v_2) \neq \{1, 2, 6\}$. Therefore, the coloring $\phi'$ can be extended to a good coloring of $G$.

Subcase 3.2: $d_G(u_1) = d_G(v_1) = 3$, $d_G(u_2) = d_G(v_2) = 2$.

Then $|A_\phi(uw)| \geq 1$ and $|A_\phi(vw)| \geq 1$. If one of $|A_\phi(uw)|$ and $|A_\phi(vw)|$ is at least 2, or $A_\phi(uw) \neq A_\phi(vw)$, then both $uw$ and $vw$ can be colored. So we may assume that $A_\phi(uw) = A_\phi(uw) = \{6\}$. Without loss of generality, we may further assume that $\phi(uu_1) = 1$, $\phi(uu_2) = 2$, $\phi(vv_1) = 3$, $\phi(vv_2) = 4$, $\phi(u_2 u_2') = \phi(v_2 v_2') = 5$.

By a similar argument used in Subcase 3.1, we deduce that $S_\phi(u_2') = \{3, 4, 5\}$ and $S_\phi(v_2') = \{1, 2, 5\}$. Then we can change the colors of $uu_2$ and $vv_2$ both to 6. Now we get a good coloring of $G$ by assigning color 2 to $uw$ and color 4 to $vw$. □

This completes our proof for Theorem 2.

## 3. Conclusions

In this paper, we present a slightly different proof of a result proved by Gu et al. [11]. Lemma 1 for forbidden colors is crucial for our proof, and it can be extended to a more general setting. For $\Delta \geq 4$, Conjecture 1 is still open. It will be interesting to consider the case $\Delta = 4$ for our future work.

**Author Contributions:** Conceptualization, L.C. and Y.L.; methodology, L.C.; validation, L.C. and Y.L.; formal analysis, Y.L.; investigation, Y.L.; resources, Y.L.; writing—original draft preparation, Y.L.; writing—review and editing, L.C.; visualization, Y.L.; supervision, L.C.; project administration, L.C.; funding acquisition, L.C. All authors have read and agreed to the published version of the manuscript.

**Funding:** This research was funded by the Natural Science Foundation of Fujian Province (No. 2020J05058) and the Fundamental Research Funds for the Central Universities of Huaqiao University (ZQN-903).

**Institutional Review Board Statement:** Not applicable.

**Informed Consent Statement:** Not applicable.

**Acknowledgments:** The authors are very grateful to R. Luo and X. Zhou for their helpful comments and suggestions. This work is supported by the Natural Science Foundation of Fujian Province (No. 2020J05058) and the Fundamental Research Funds for the Central Universities of Huaqiao University (ZQN-903).

**Conflicts of Interest:** The authors declare no conflict of interest.

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
