# Peer review of "A New Proof for a Result on the Inclusion Chromatic Index of Subcubic Graphs"

_axioms, doi:10.3390/axioms11010033_

Round 1
Reviewer 1 Report
Please see comments in attached report.

Reviewer 2 Report
Review of A New Proof For A Result On The Inclusion Chromatic Index of Subcubic Graphs
The authors provide a “new proof” for a result on the chromatic index of subcubic graphs in terms of an inequality where for the complete bipartite graph, K2,3 the chromatic index becomes exactly 7. The latter result also follows from a previous theorem. While the result and proof are both interesting, the manuscript needs to bring the work in a broader context to increase the impact of the paper. Furthermore, the paper lacks novelty and it seems too narrow in focus as a single result of an inequality on a specialized set of graphs does not appeal to a broader audience. I strongly suggest the authors to expand the scope to include some more generalized results/ideas on chromatic indices/polynomials.
The authors must expand their introduction by pointing out broader applications of the chromatic index and chromatic polynomials. In order to help the authors, the DOIs of a few highly relevant references are provided below. The authors should also look up the back references contained in the references mentioned below which should help to bring the authors’ work into a broader context and this significantly increase the impact of the authors’ work.
The authors give only 6 references with two of them by Zhang and co-workers. There is a vast literature on chromatic polys of graphs with applications to many fields including parallel architecture and stat mech- all of which appears to have been neglected.
The proof provided by the authors seems to be way too complex than the result as the authors have to decide this part into so many subcases. Thus it becomes difficult to follow the logic and I strongly urge the authors to see if they can recast this section- which will again increase the impact of the paper.
The authors should make some comments about any extensions of their results for example to hypercubes as colorings of hypercubes have numerous applications (see below). There are also chemical applications to the chromatic index (nearest neighbour exclusion isomers (see refs below). I strongly urge the authors to point out these applications of colorings to substantially increase the impact of their work.
The authors have not provided a conclusion section which could include their main finding and any potential future works on related problems, especially any unsolved problems that could stimulate further research on the topic.
Relevant references (DOIS/refs) that must be included.
- Motoyama and H. Hosoya, J. Math. Phys., 18, 1485 (1977).
- Montroll, in Applied Combinatorial Mathematics, E. F. Beckenbach, Ed., Wiley, New York, 1964.
- Lichtman and R. B. McQuistan, J. Math. Phys., 124, 1258 (1973)
- DOI: 1002/jcc.540060513
- 1002/jcc.26118
- 1002/jcc.540060207
- NEW YORK ACAD. SCI.; USA; DA. 1979(05), VOL. 319; PP. 33-36
- 1016/j.parco.2012.07.001
Recommendation: The paper needs to undergo major revision and reviewed again.
Author Response
1. We added the relevant references as the referees' suggest.
2.We rewrote the proof for the $1$-thread case by seperating Claim 1 out.
3. We added a conclusion section.
4. We added some comments about some extensions of the results and added some applications of colorings in the introduction.
Round 2
Reviewer 1 Report
I think the revised paper is fine and suitable for publication.
Reviewer 2 Report
The revised manuscript is significantly improved with respect to the content and broader appeal but it will be benefited by improvement in English. Perhaps the authors should get the manuscript proof-read by native speakers or by MDPI's language expertise.